# Phylogenetic Analysis of Pyruvate-Ferredoxin Oxidoreductase, a Redox Enzyme Involved in the Pharmacological Activation of Nitro-Based Prodrugs in Bacteria and Protozoa

**DOI:** 10.3390/biology13030178

**Published:** 2024-03-09

**Authors:** Seth Duwor, Daniela Brites, Pascal Mäser

**Affiliations:** 1Swiss Tropical and Public Health Institute, 4123 Allschwil, Switzerland; 2Faculty of Science, University of Basel, 4001 Basel, Switzerland; 3Department of Clinical Pharmacology and Toxicology, University Hospital Zurich, University of Zurich, 8091 Zurich, Switzerland

**Keywords:** pyruvate-ferredoxin oxidoreductase, metronidazole, reductive bioactivation, antimicrobial spectrum, comparative genomics, horizontal gene transfer

## Abstract

**Simple Summary:**

The distribution of typical bacterial redox enzymes such as pyruvate-ferredoxin oxidoreductase (PFOR) in protozoa remains interestingly puzzling. Previous studies have demonstrated diverse cellular localizations of PFOR in some amitochondriate anaerobic protozoa. PFOR is of particular pharmacological importance because it catalyzes the reductive bio-activation of nitro-based prodrugs to cytotoxic radical metabolites. Metronidazole was developed primarily as an antiprotozoal agent against infections caused by *Trichomonas vaginalis*. However, its antimicrobial spectrum was subsequently expanded to cover anaerobic bacterial infections. It has been shown that mutations in the genes encoding PFOR result in the resistance of PFOR-possessing anaerobic protozoa and bacteria to nitro-based prodrugs. Deciphering the evolutionary history of PFOR is crucial for deepening our understanding of the evolution of anaerobic pathogens and unfolding new approaches for drug discovery and targeting in pathogen chemotherapy.

**Abstract:**

The present frontrunners in the chemotherapy of infections caused by protozoa are nitro-based prodrugs that are selectively activated by PFOR-mediated redox reactions. This study seeks to analyze the distribution of PFOR in selected protozoa and bacteria by applying comparative genomics to test the hypothesis that PFOR in eukaryotes was acquired through horizontal gene transfer (HGT) from bacteria. Furthermore, to identify other putatively acquired genes, proteome-wide and gene enrichment analyses were used. A plausible explanation for the patchy occurrence of PFOR in protozoa is based on the hypothesis that bacteria are potential sources of genes that enhance the adaptation of protozoa in hostile environments. Comparative genomics of *Entamoeba histolytica* and the putative gene donor, *Desulfovibrio vulgaris*, identified eleven candidate genes for HGT involved in intermediary metabolism. If these results can be reproduced in other PFOR-possessing protozoa, it would provide more validated evidence to support the horizontal transfer of *pfor* from bacteria.

## 1. Introduction

The oxidative decarboxylation of pyruvate to acetyl-CoA and CO_2_ is a central biochemical reaction that links glycolysis to the tricarboxylic acid cycle. Under aerobic conditions, it is catalyzed by the pyruvate dehydrogenase complex [1]. In most anaerobic and microaerophilic microorganisms, however, pyruvate is decarboxylated by pyruvate: ferredoxin oxidoreductase (PFOR; E.C. 1.2.7.1) [2,3]. PFOR is an oxygen-sensitive iron-sulfur protein of the 2-oxo acid ferredoxin oxidoreductase (OFOR) superfamily [4]. The fermentative decarboxylation of pyruvate by PFOR generates electrons that are transferred to either ferredoxin or flavodoxin [5,6]. Due to the reversibility of this reaction, PFOR is also called pyruvate synthase. PFOR is of high interest to (at least) three different research areas: (i) biochemistry as a means for oxidation of pyruvate under anaerobic conditions; (ii) pharmacology because it can reduce and thereby activate nitro-based prodrugs such as metronidazole; and (iii) evolutionary biology as it is an ancient enzyme that stems from the times when life on earth was anaerobic.

PFOR orthologs are present in the archaea, many anaerobic bacteria, and some anaerobic eukaryotes [7]. In amitochondriate protists, the enzyme has varying localizations [2,3]. Based on the localization of PFOR, amitochondriate protists can be classified as type 1 or type 2 protists [8]. In type 1 protists, such as *Giardia duodenalis*, *Entamoeba histolytica* and *Cryptosporidium parvum*, PFOR is localized in the cytosol [9]. The PFOR in type 2 protists is localized in the hydrogenosome, as observed in the trichomonads [10,11]. The hydrogenosome is a double-membrane-bounded organelle [12,13] that shares common ancestry with mitochondria [14,15,16]. It is a powerhouse for the fermentative decarboxylation of pyruvate to hydrogen, carbon dioxide, and ATP in the absence of molecular oxygen [17,18,19]. Interestingly, the free-living mitochondriate algae *Chlamydomonas reinhardtii* is able to switch from aerobic to anaerobic metabolism by upregulating the expression of PFOR, which localizes to and functions in the chloroplastic stroma [20,21]. However, the N-terminus of PFOR in the facultative anaerobic, photosynthetic protist *Euglena gracilis* localizes to the mitochondrion [22]. Due to the patchy occurrence and diverse localizations of PFOR in eukaryotes, bacterial origins have been proposed [8,22,23]. One plausible mechanism to explain this phenomenon is horizontal gene transfer (HGT).

The role of HGT in the acquisition of virulence and multiple drug resistance genes within prokaryotes is well established [24,25,26]. However, its involvement in the transfer of genetic material from prokaryotes to eukaryotes or within eukaryotes remains controversial [19,20,21,22,23,24,25,26,27]. HGT was proposed to be an essential evolutionary mechanism by which unicellular eukaryotes acquire functional genetic material from bacteria that reside in close proximity, such as in the host gastrointestinal or genito-urinary tract [28,29,30,31]. It was also proposed that some protozoan parasites residing in the same host environment as bacteria can receive foreign DNA fragments through phagocytic feeding of bacteria [32]. The acquisition of exogenous genetic material may facilitate the adaptation, pathogenicity, and survival of protozoa in the face of strong selective pressures [30,33,34,35,36,37]. Therefore, protozoa may have acquired genes encoding bacterial redox enzymes such as PFOR via HGT, which facilitated their survival in deteriorating environments [38,39,40]. A previous study that utilized phylogenetic methods on the whole genome sequences of *E. histolytica* and *T. vaginalis* inferred 68 and 153 cases of HGT, respectively. Most of the transferred genes were found to encode enzymes involved in prokaryote intermediary metabolism, and the study concluded that prokaryotes that reside in close proximity to *E. histolytica* and *T. vaginalis* were potential gene donors [41].

The presence of PFOR (Figure 1) sensitizes anaerobic microorganisms to nitro-based antimicrobials such as metronidazole or tinidazole. These nitro-heterocyclic and nitro-aromatic compounds are prodrugs that need to be activated by chemical reduction of the nitro group, which leads to the formation of potent nitro radicals. In analogy to Paul Ehrlich’s concept of the magic bullet, such drugs can be thought of as magic bombs that need to be triggered by electron transfer [42].

Metronidazole is a synthetic nitro-imidazole derivative effective against most obligate anaerobic bacteria and protozoan parasites such as *G. duodenalis*, *E. histolytica*, and *T. vaginalis* [43,44]. It exerts its antimicrobial effect by interfering with nucleic acid synthesis, resulting in DNA damage in the target pathogens [45]. Electrons from the initial redox reaction catalyzed by PFOR are transferred to ferredoxin, which has a sufficiently low redox potential to reductively activate metronidazole to its toxic radical metabolites. Through a series of electron transfer reactions, different radical intermediates are formed that act primarily as alkylating agents that disrupt DNA synthesis and the growth of invading pathogens [46]. *G. duodenalis* and *E. histolytica* strains with decreased or absent expression of *pfor* exhibited increased metronidazole resistance [47]. However, some recent studies suggest the involvement of additional enzymes that also play central roles in the activation of and resistance to metronidazole [48,49,50,51]. For example, *T. vaginalis* has been shown in some studies to display a significant resistance to metronidazole only after both PFOR and NAD-dependent malic enzymes are inactivated in the hydrogenosome [52,53].

Here we analyze the distribution of PFOR in selected protozoan parasites, re-investigating the question of whether the genes have been acquired from prokaryotes by horizontal transfer and, if so, which other genes came along with PFOR.

## 2. Materials and Methods

### 2.1. Protein Sequences

The protein sequences were retrieved from the UniProt protein database (www.uniprot.org, accessed on 20 September 2022). In addition to PFOR, three eukaryote ‘housekeeping’ proteins were included as controls: glyceraldehayde-3-phosphate dehydrogenase (GAPDH) (EC 1.2.1.12), tubulin alpha chain (TUBa), and DNA-directed RNA polymerase II subunit beta I (RPB1) (EC 2.7.7.6). See Appendix A for accession numbers.

### 2.2. Sequence Alignment

Sequence similarity queries were performed using the proteins of interest against the NCBI non-redundant protein sequence database with the basic local alignment search tool program (BLASTP version 2.12.0+) from the National Center for Biotechnology Information (NCBI) website (https://blast.ncbi.nlm.nih.gov/Blast, accessed on 23 September 2022) [54]. Initially, the queries were performed using the control proteins of each parasite. The same procedure was performed for the corresponding redox enzymes using an exploratory BLAST “pilot approach” with the following algorithm parameters: expected threshold: 1 × 10^−15^, word size: 6, matrix: bLOSUM62, gap costs: existence 11, extension 1 and compositional adjustments: conditional compositional score matrix adjustments [55]. Proteome-wide blastp searches were run locally on a Linux PC, and the results, in tabular format, were parsed with self-made Perl scripts.

### 2.3. Phylogenetic Analyses

Multiple sequence alignments and phylogenetic trees were made with the Molecular Evolutionary Genetics Analysis (MEGA) software, version 10 using the muscle algorithm with default parameters [56], followed by manual trimming of the loose, unaligned ends of the alignment. The evolutionary distances were computed using the Jones-Taylor-Thornton (JTT) substitution model. These phylogenetic trees were constructed using the neighbor-joining algorithm [57]. Bootstrapping was performed with 1000 rounds of replication [58].

### 2.4. Screening Selected Proteomes against HMM Profile Libraries

Multiple sequence alignments were converted to position-dependent scoring matrices using the command *hmmbuild* of the HMMer 3.0 package. The resulting profiles were concatenated and converted to HMM profile libraries using *hmmpress*. Complete proteomes of representative organisms from each eukaryotic supergroup and selected bacteria were downloaded from the ensemblgenomes database (ftp://ftp.ensemblgenomes.org/pub/, accessed on 20 September 2022). The HMM profile libraries were used to screen the downloaded proteomes with *hmmscan* of the HMMer 3.0 package. All the steps were executed with self-made Perl scripts on a Linux computer.

### 2.5. Gene Enrichment Analysis

The gene IDs of the proteins that were exclusively present in the selected protozoan species and *D. vulgaris* were transferred to the Gene Ontology (GO) website (http://geneontology.org/, accessed on 29 September 2022) for gene enrichment analysis. The analyses were performed using the PANTHER Overrepresentation Test (released on 13 October 2022) and the GO database https://doi.org/10.5281/zenodo.6799722, accessed on 29 September 2022 (PANTHER version 17.0, released on 1 July 2022). The PANTHER Pathway for biological processes was used for the annotation data set, and Fisher’s exact test with false discovery rate (FDR) correction was used for the statistical analysis. Only results with *p*-values < 0.05 were included. 

## 3. Results

### 3.1. BLAST Pilot Experiment

In a first exploratory attempt towards understanding the phylogeny of PFOR, we compared its performance to that of eukaryotic housekeeping proteins when used as a query in blastp searches. The searches were performed online at NCBI, profiting from the ‘organism’ feature that allows to in- or exclude particular clades of the tree of life. We would expect the number of hits returned for a given query to rise with increasing search space, e.g., when moving up from the level of genus to a higher taxonomic order. This was the case for the control proteins glyceraldehayde-3-phosphate dehydrogenase (GAPDH), RNA polymerase II subunit B1 (RPB1), and α-tubulin (TUBa; Figure 2, grays); however, it was not evident for PFOR (Figure 2, red). For instance, *E. histolytica* PFOR returned hits from three *Entamoeba* species other than *E. histolytica* (i.e., *E. dispar*, *E. invadens*, and *E. nutalli*), the same as for the control proteins. Increasing the search space to include all amoebozoa produced hits only from three additional species (*Acanthamoeba castellanii*, *Mastigamoeba balamuthi*, and *Pelomyxa schiedti*) with PFOR, but clearly more with the control proteins. Similarly, only thirteen micromonads other than *Giardia* spp. possessed a PFOR ortholog, and only one apicomplexan (*Porospora gigantea*, a gregarine intestinal parasite of lobsters) other than *Cryptosporidium* spp. (Figure 2). While this simple approach may be biased due to differences in coverage and the number of sequences submitted to GenBank, it nevertheless showed that PFOR in the amoebozoa, micromonada, and apicomplexa does not occur ubiquitously but punctually, only in particular genera.

### 3.2. Distribution of PFOR and Control Proteins along the Tree of Life 

To explore the distribution of PFOR orthologs in an unbiased way, we had to concentrate on species for which whole genome sequence data were available. We constructed hidden Markov model-based profiles for the different query proteins, i.e., PFOR, GAPDH, RPB1, and TUBa. These profiles were then implemented for genome-wide surveys, searching the predicted proteomes of different eukaryotes. The score of the best hit was noted for each species, and the scores for each profile were normalized to a maximum value of 1 for the best overall hit (to account for the fact that longer profiles return a higher maximal score than shorter ones). This survey confirmed an overall scarcity of PFOR orthologues in eukaryotes, with an apparent absence from animals, land plants, euglenozoa, and fungi, a very punctual distribution in the alveolates, stramenopiles, amoebozoa, and chlorophytes, and a wider occurrence in all the metamonades for which predicted proteomes were available (Table 1). There were only two rhizarian species with available proteome sequences in the ENSEMBL genome database, neither of which had a hit for PFOR (Table 1). All the species in the genus *Cryptosporidium* returned hits for PFOR. However, the closely related parasites in the genera *Eimeria* and *Cyclospora* did not. The presence of a strong PFOR hit in *Blastocystis hominis* was surprising and, to the best of our knowledge, has not been reported before. A further surprise was the presence of PFOR hits in the nematodes *Necator americanus* and *Trichuris trichiura*, gastrointestinal human parasites. However, upon reciprocal blastp searches, the two hits (GenBank XP_013306304 and CDW60499) turned out to be identical to PFOR sequences from bacteria (PVY31832 and WP_000628243, respectively). Hence, these sequences could also result from a contamination of nematode DNA with bacterial DNA and were therefore not included for further analysis. 

### 3.3. Phylogenetic Tree of PFOR

To investigate the evolutionary history of eukaryotic PFOR further, we drew a phylogenetic tree of the PFOR sequences from eukaryotes identified with profile searches and the blastp approach, supplemented with the most closely related sequences from prokaryotes as identified by blastp searches using the eukaryotic PFOR orthologs as queries. The hierarchy of the major branches of the PFOR tree did not unequivocally resolve, as indicated by the bootstrap numbers below 60% (Figure 3). Nevertheless, it was evident that the PFOR sequences clustered according to their phylogeny: the sequences from the amoebozoa, apicomplexa, and green algae built their own clades, each with a high bootstrap support (Figure 3). Among the sequences from metamonads, however, those from intestinal parasites of vertebrates formed their own clade, separate from that of the free-living *Paratrimastix pyriformis* and the termite endosymbiont *Streblomastix strix*. Eukaryote and prokaryote sequences did not intermix: all the bacterial sequences ended up in the same branch, with a bootstrap support of 100% (Figure 3).

### 3.4. Proteome-Wide BLAST Surveys

The patchy distribution of PFOR orthologues in eukaryotes (Table 1, Figure 2) is in agreement with the hypothesis that PFOR was obtained by eukaryotes via horizontal gene transfer from prokaryotes. To further explore this hypothesis, we asked the question whether—if PFOR was indeed acquired horizontally—any other bacterial genes came along with it. We first addressed this question using PFOR from *E. histolytica*. Its most similar ortholog from prokaryotes, as determined by blastp, is the one from *Desulfovibrio vulgaris*, a Gram-negative anaerobic bacterium that occurs in the environment and in the mammalian gut. Hence, we conducted a proteome-wide BLAST survey, searching every protein of *E. histolytica* (*n* = 7455) against the proteome of *D. vulgaris* (*n* = 3880). In parallel, we performed the same search of all *E. histolytica* proteins against the proteome of *Dictyostelium discoideum* (*n* = 13,233), a free-living amoeba that lacks PFOR (Table 1). *Entamoeba* is much more closely related to *Dictyostelium* than to *Desulfovibrio*; furthermore, *Dicytostelium* has a larger proteome than *Desulfovibrio*. So clearly, we would expect *Entamoeba* proteins, in general, to return a higher-scoring best hit from *Dictyostelium* than from *Desulfovibrio*. This was indeed the case: the majority of points deviated from the diagonal towards the lower right corner of the 2D plot (Figure 4). Of the 7455 *E. histolytica* proteins, 3310 (44%) did not return a hit from either *D. discoideum* or *D. vulgaris* (zero point of the 2D plot). PFOR did not have a hit in *D. discoideum*, but it was the highest scoring of all *E. histolytica* proteins in *D. vulgaris* (Figure 4). Interestingly, several other E. histolytica proteins also had a good-scoring blastp hit in *Desulfovibrio* but not in *Dictyostelium* (Figure 4).

A similar picture was obtained when the same approach was carried out with *C. parvum* (Figure 5). Here, we used *Eimeria tenella* (*n* = 8599 proteins) as a related apicoplastid parasite that does not possess PFOR (Table 1). As with *E. histolytica*, 44% of the *C. parvum* proteins (1689 of *n* = 3805 total proteins) did not return a blastp hit from either proteome. The remainder of the *C. parvum* proteins showed a distribution that was, as expected, clearly skewed towards the lower right corner, i.e., towards higher similarity to *Eimeria* than to *Desulfovibrio* (Figure 5). Again, *C. parvum* PFOR stood out as the protein with the highest scoring hit in *Desulfovibrio*, followed by a couple of other *C. parvum* proteins at the ordinate of the 2D diagram (Figure 5).

### 3.5. GO Term Enrichment Analysis

Concentrating on the subset of proteins from anaerobic protozoa that, like PFOR, had a higher similarity to bacterial than to eukaryotic proteins, we investigated what—if anything—this subset of sequences had in common. For this purpose, we performed a gene ontology (GO) term enrichment analysis, profiting from the good level of annotation of the *E. histolytica* proteome [59,60]. Biological Process was used as the annotation data category to analyze the gene list corresponding to the 40 *E. histolytica* proteins of highest similarity to *D. vulgaris* but not to *D. discoideum*. The enrichment of GO terms was calculated with reference to the complete set of 7959 *E. histolytica* genes. Nine of the 40 query genes were unmapped; hence, the remaining 31 genes were used for the enrichment analysis. The significantly enriched biological processes are summarized in Table 2. Pyruvate metabolism was overrepresented by a factor of 40, followed by related but less specific categories. The general GO categories overrepresented in the set of 31 *E. histolytica* genes also dealt with energy metabolism, i.e., the carbohydrate metabolic process and pathways involved in the generation of precursor metabolites and energy (Table 2). Interestingly, the list of genes with mapped GO terms that were enriched included pyruvate phosphate dikinase (XP_657332), the enzyme that converts pyruvate to phosphoenolpyruvate in a reaction typical for C4 plants. The surprising presence of pyruvate phosphate dikinase in *E. histolytica* has been noted before [61]. The enrichment of GO terms in the subset of *E. histolytica* genes whose products have a higher resemblance to bacterial rather than to eukaryotic proteins suggests that these genes do not comprise a random selection. Rather, they share a common purpose related to the metabolism of pyruvate. 

## 4. Discussion

The distribution of PFOR compared to that of GAPDH along the taxonomic tree of the selected protozoan species confirmed the relatively limited occurrence of PFOR in eukaryotes. The BLAST results further confirmed that the distribution of PFOR is highly restricted to a few organisms in the eukaryotic domain, which is potentially interesting and worth looking at in closer detail. Due to the absence of deviation observed in the total number of hits for PFOR compared to GAPDH in the genera *Giardia*, *Entamoeba,* and *Cryptosporidium*, it can be inferred that all species in these genera possess PFOR. The expanded cladograms of *Entamoeba* and *Cryptosporidium* derived from the whole proteome analysis, confirmed the findings from the BLAST results, since all the selected species in these genera with available whole proteome sequences from the ensemblgenomes database had positive hits for PFOR. Taking a broader view of all the selected protozoan species in the alveolate clade, only *C. parvum* showed a positive hit for PFOR. This is an interesting observation in support of a possible horizontal acquisition of *pfor* because the expanded cladogram of the genus *Cryptosporidium* showed that the closely related species in the genera *Eimeria* and *Cyclospora* had no hits for PFOR. A similar pattern observed in the expanded cladogram of all the available species of the genus *Entamoeba* and those of the related genera *Dictyostelium*, *Cavenderia*, *Tiegemostilium,* and *Planoprotostelium*, further substantiated the hypothesis that *pfor* was most likely horizontally acquired. However, due to the limited availability of additional species with whole proteome sequences in the metamonad clade, it was not possible to generate an expanded cladogram for the genus *Giardia* and its closely related genera to support the horizontal transfer hypothesis of *pfor* in *G. duodenalis.* Consequently, all further analyses to support the hypothesis were limited to *C. parvum* and *E. histolytica.*


The phylogenic tree generated from the protein sequences of PFOR using the neighbor-joining method showed some degree of incongruence with the expected phylogeny of eukaryotes. Yet the statistical support for the most inner nodes was generally weak, indicating the strong divergence of PFOR sequences between distantly related groups. This phylogenetic incongruence could indicate multiple sources of the acquisition of *pfor* in protozoa; however, the bacterial and protozoan species were clearly separated into distinct clades. This pattern is in agreement with a single horizontal transfer event from bacteria to eukaryotes. 

It is striking that PFOR in eukaryotes occurs in gut endosymbionts or endoparasites that live inside their host in close contact with bacteria. This suggests two alternative hypotheses: (1) the proximity to bacteria facilitated the acquisition of *PFOR* by horizontal gene transfer; or (2) the anaerobic environment enforced maintenance of an ancestral *PFOR* gene, whereas free-living protists have lost the gene. The two scenarios are not necessarily mutually exclusive, and both are in agreement with the observed phylogenetic tree. However, the fact that there are groups of anaerobic eukaryotes that do not possess *PFOR*, such as trypanosomatid parasites of insects, may speak against the second hypothesis. 

In view of the proximity hypothesis that supports the horizontal transfer of genes from bacteria, the bacterium with the highest normalized similarity score of 1.0 for PFOR, *D. vulgaris*, qualifies as the most suitable model of the putative gene donor or representative of an extant bacterial lineage of gene donors. *D. vulgaris* is a Gram-negative, anaerobic, non-spore-forming, curved rod-shaped, sulfate-reducing bacterium capable of producing hydrogen sulfide in the host gastrointestinal tract and, as such, dwells in close proximity with some invading protozoan parasites. The comparative whole proteome BLAST analysis of *E. histolytica* and *C. parvum* against their corresponding related control species, *D. discoideum* and *E. tenella*, and the bacterium *D. vulgaris* demonstrated significantly high hits for some proteins that are exclusively present in *D. vulgaris* but absent in the more closely related protozoan control species. 

Out of the forty selected genes in *E. histolytica* and *D. vulgaris*, eleven genes predicted to be involved in biological processes related to small molecule metabolism, the generation of precursor metabolites, and carbohydrate metabolism were significantly overrepresented. The pyruvate metabolic process is a biological process related to the small molecule metabolic pathway, under which three out of the forty exclusively expressed proteins are functionally categorized. The findings suggest that the eleven genes that were significantly enriched did not occur by chance compared to the reference genome of *E. histolytica*. Furthermore, the small *p*-values indicate that the outcome of the gene enrichment analysis was non-random and worth looking at in further detail.

The significantly enriched genes can be analyzed further using syntenic approaches to give substantial information on how they are structurally related according to their relative position at the chromosomal level. If it can be established that these genes are located close to each other at the chromosomal level, this would provide stronger evidence in support of the hypothesis that they were most likely horizontally transferred together with *pfor* as a genetic fragment from bacteria. Moreover, if these findings can be reproduced in other PFOR-possessing protozoan species, this will provide more evidence in support of the horizontal gene transfer hypothesis. 

## 5. Conclusions

The redox enzyme PFOR plays significant roles in the energy metabolism and growth of pathogens that dwell predominantly in oxygen-deprived niches. From the BLAST pilot analysis as well as the whole proteome analysis, it can be inferred that the presence of PFOR is limited to a few protozoan species in the eukaryotic domain, whereas a relatively large proportion of prokaryotes possess PFOR. Protozoa belonging to the genera *Cryptosporidium*, *Giardia*, *Entamoeba,* and *Blastocystis* are anaerobic parasites that reside predominantly in close proximity with bacterial species in the mammalian host’s gastrointestinal tract [62], whereas those belonging to the genera *Trichomonas* and *Tritrichomonas* are sexually transmitted extracellular anaerobic parasites that also dwell in close proximity with bacteria in the host genito-urinary tract [63,64]. Appendix A illustrates how the lifecycles of some selected protozoa support the horizontal acquisition of PFOR from bacteria based on the proximity hypothesis.A plausible explanation for the restricted occurrence of PFOR in the above-mentioned protozoa is based on the hypothesis that bacterial species serve as potential sources for the acquisition of genes that enhance the optimal adaptation of these protozoa in hostile host environments. The expanded cladograms of *Entamoeba* and *Cryptosporidium,* with their closely related genera, substantiated this hypothesis. Although the phylogeny based on the protein sequences of PFOR demonstrated some degree of phylogenetic incongruence, the PFOR of none of the protozoa was monophyletic with that of bacteria. The observed monophyletic relationship of the PFOR of *E. invadens* with *Giardia* species and that of *E. gracilis* with Cryptosporidium species, is suggestive that these two distinct groups of protozoa may have acquired *pfor* from a common ancestral lineage. The exclusively expressed proteins obtained from *E. histolytica* and the putative bacterial gene donor, *D. vulgaris*, showed an over-representation of eleven genes involved in small-molecule metabolism, the generation of precursor metabolites, and carbohydrate metabolism. If these results obtained from *E. histolytica* and *D. vulgaris* can be reproduced in other PFOR-possessing protozoan species, it would provide more evidence to support the horizontal transfer of *pfor* from bacteria. Subsequent syntenic analyses of the significantly enriched genes would be required to provide further information regarding the positional relatedness of these genes at the chromosomal level. 

## Figures and Tables

**Figure 1 biology-13-00178-f001:**
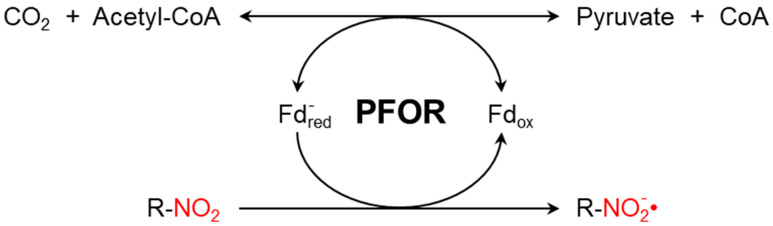
PFOR-mediated fermentative decarboxylation of pyruvate (reversible) and reductive activation of nitro-based prodrugs (irreversible due to a cascade of redox reactions converting the nitroso radical to hydroxylamine).

**Figure 2 biology-13-00178-f002:**
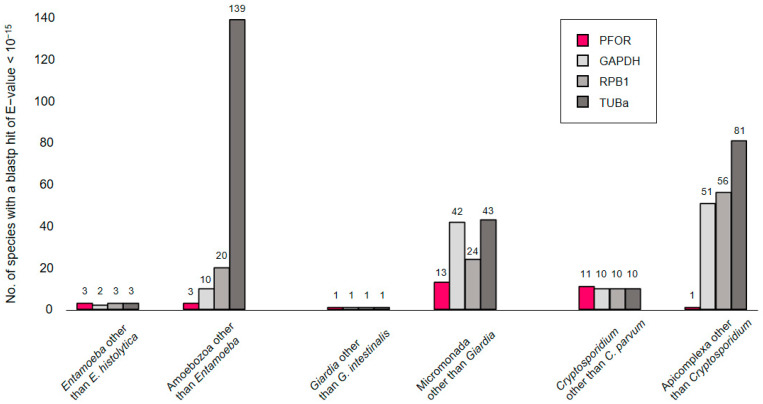
Number of blastp hits for PFOR (red) and control proteins (gray) in different search spaces. Blastp searches were performed online at NCBI using default parameters and an E-value cut-off of 10^−15^. While PFOR and the control proteins performed equally well at the level of selected genera, PFOR clearly returned fewer hits than the control sequences when expanding the search space by moving up in taxonomic hierarchy.

**Figure 3 biology-13-00178-f003:**
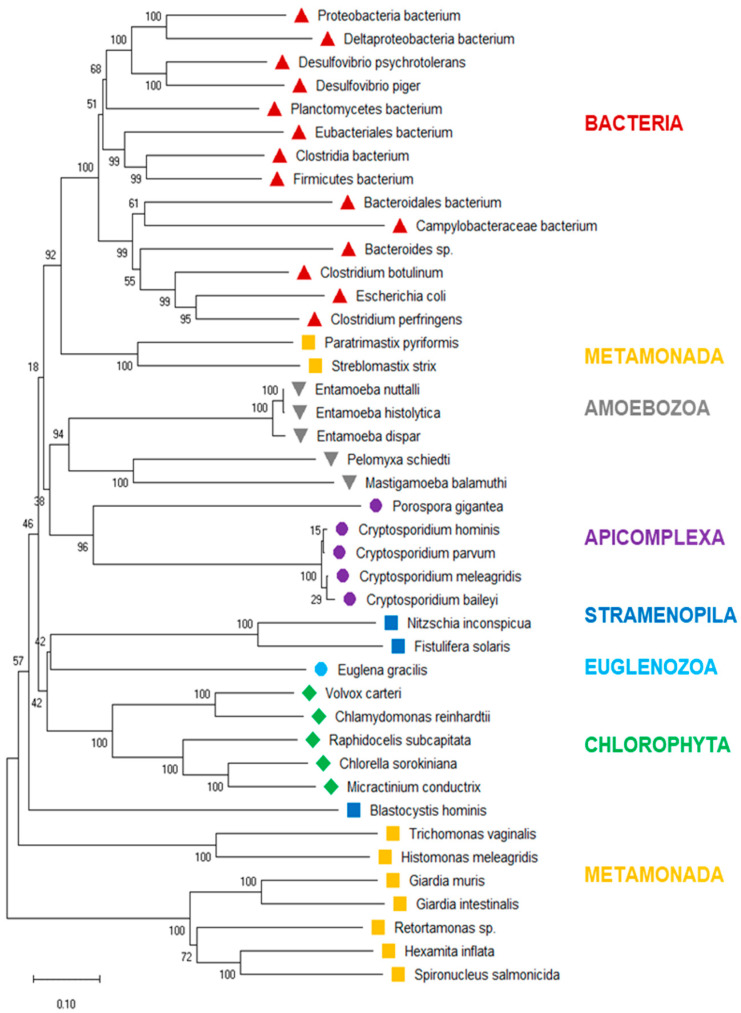
Neighbor-joining tree of eukaryote PFOR orthologs supplemented with the most similar sequences from prokaryotes. The scale bar indicates the number of changes per site. The numbers at the branch sites indicate percent positives of 1000 rounds of bootstrapping. Major phylogenetic lineages are indicated in color.

**Figure 4 biology-13-00178-f004:**
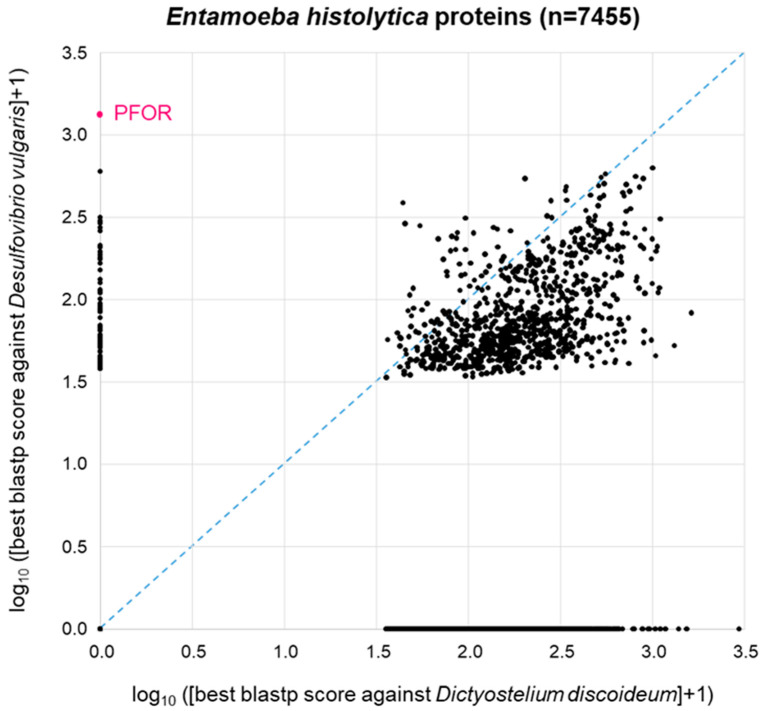
BLAST survey of all proteins from *E. histolytica* against the proteomes of the related amoeba *D. discoideum* and the bacterium *D. vulgaris*. For each *E. histolytica* protein, the score of the best blastp hit from each proteome is plotted. Proteins at the zero point did not return a hit from either proteome.

**Figure 5 biology-13-00178-f005:**
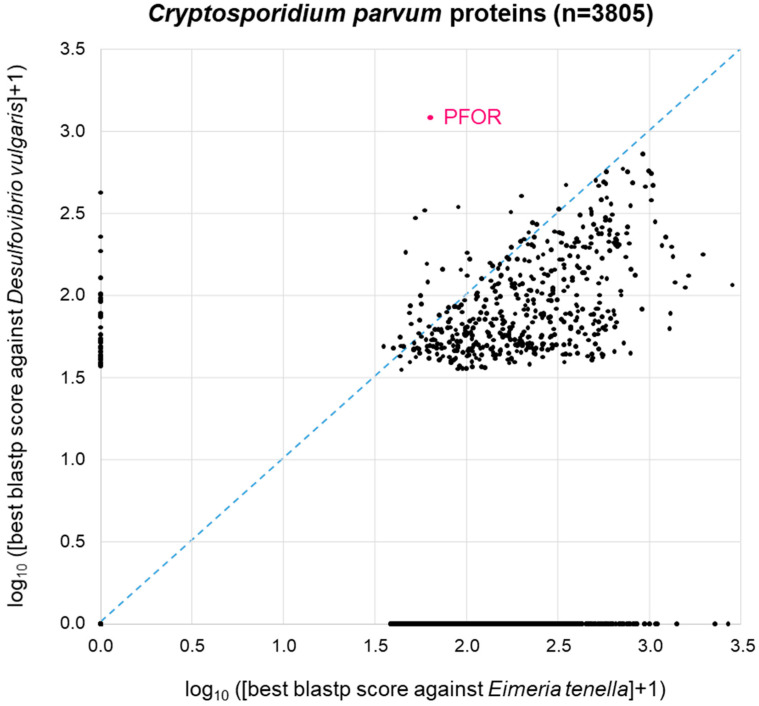
BLAST survey of all proteins from *C. parvum* against the proteomes of the related apicoplastid *E. tenella* and the bacterium *D. vulgaris*. For each *C. parvum* protein, the score of the best blastp hit from each proteome is plotted. Proteins at the zero point did not return a hit from either proteome.

**Table 1 biology-13-00178-t001:** Searching the predicted proteomes of completely sequenced eukaryotes for the presence of PFOR and control proteins with profile hidden Markov models. Normalized HMMer best scores are shown per proteome. Only a selection of those proteomes negative for PFOR are included.

Taxon	Species	PFOR	GAPDH	TUBa	RPB1
Stramenopiles	*Blastocystis hominis*	0.89	0.91	0.82	0.59
	*Phytophthora infestans*	0.22	0.97	0.99	0.94
	*Albugo laibachii*	0.01	0.95	0.97	0.92
	*Ectocarpus siliculosus*	0.01	0.93	0.97	0.85
Alveolates	*Plasmodium falciparum*	0.01	0.92	1.00	1.00
	*Cryptosporidium parvum*	0.98	0.97	0.97	0.91
	*Toxoplasma gondii*	0.01	0.94	0.99	0.89
	*Tetrahymena thermophila*	0.01	0.89	0.99	0.59
Rhizaria	*Plasmodiophora brassicae*	0.01	0.84	1.00	0.87
	*Reticulomyxa filosa*	0.01	0.57	0.93	0.73
Euglenozoa	*Trypanosoma brucei*	0.01	0.95	0.98	0.72
	*Phytomonas serpens*	0.00	0.89	0.97	0.69
	*Naegleria gruberi*	0.01	0.91	0.99	0.85
Amoebozoa	*Entamoeba invadens*	0.96	0.92	0.79	0.66
	*Entamoeba histolytica*	0.99	0.97	0.47	0.67
	*Dictyostelium purpureum*	0.01	0.93	0.82	0.93
	*Dictyostelium discoideum*	0.01	0.92	0.83	0.93
Metamonada	*Giardia lamblia*	0.80	0.91	0.98	0.62
	*Spironucleus salmonicida*	0.78	0.83	0.92	0.58
	*Trichomonas vaginalis*	0.88	0.70	0.97	0.70
Animalia	*Caenorhabditis elegans*	0.01	0.98	0.97	0.93
	*Schistosoma mansoni*	0.01	0.93	1.00	0.82
	*Drosophila melanogaster*	0.01	0.95	1.00	0.93
Fungi	*Sporisorium reilianum*	0.16	0.96	0.87	0.86
	*Saccharomyces cerevisiae*	0.11	0.94	0.89	0.88
	*Aspergillus fumigatus*	0.17	0.95	0.93	0.88
Embryophyta	*Asparagus officinalis*	0.01	0.68	1.00	0.97
	*Oryza sativa*	0.01	0.97	0.98	0.92
	*Pistacia vera*	0.01	0.97	1.00	0.94
Chlorophyta	*Chlamydomonas reinhardtii*	0.98	0.92	0.99	0.86
	*Ostreococcus lucimarinus*	0.01	0.68	0.98	0.87
	*Chlorella pyrenoidosa*	0.90	0.96	1.00	0.19
Rhodophyta	*Chondrus crispus*	0.01	0.94	0.86	0.80
	*Galdieria sulphuraria*	0.01	0.95	0.94	0.52

**Table 2 biology-13-00178-t002:** GO biological processes that are enriched in the set of *E. histolytica* genes whose products have a higher similarity to *D. vulgaris* than to *D. discoideum* proteins (Figure 4). Only GO terms are shown with a *p*-value below 0.05 as determined with Fisher’s exact test and a false discovery rate (FDR) below 0.05 as calculated with the Benjamini–Hochberg procedure.

Biological Process	Fold Enrichment	*p*-Value	FDR
Pyruvate metabolic process	40	7.75 × 10^−5^	3.35 × 10^−2^
Monocarboxylic acid metabolic process	18	8.29 × 10^−5^	3.07 × 10^−2^
Carboxylic acid metabolic process	11	9.09 × 10^−5^	2.94 × 10^−2^
Oxoacid metabolic process	15	4.39 × 10^−7^	1.14 × 10^−3^
Organic acid metabolic process	15	4.63 × 10^−7^	5.99 × 10^−4^
Small molecule metabolic process	7.1	5.03 × 10^−5^	2.61 × 10^−2^
Generation of precursor metabolites and energy	24	2.94 × 10^−5^	2.54 × 10^−2^
Carbohydrate metabolic process	13	4.61 × 10^−5^	2.99 × 10^−2^

## Data Availability

The raw data supporting the conclusions of this article will be made available by the authors on request.

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
