# Peer review of "Phylogenetic Analysis of Pyruvate-Ferredoxin Oxidoreductase, a Redox Enzyme Involved in the Pharmacological Activation of Nitro-Based Prodrugs in Bacteria and Protozoa"

_biology, 2024, doi:10.3390/biology13030178_

Round 1

Reviewer 1 Report

Comments and Suggestions for Authors

The phylogenetic and proteomic analyses of PFOR described in this manuscript are an interesting topic. However, the scientific issues of the article are unfocused and the exposition is confusing. In addition, there are questions that need to be answered by the authors.

1 Whether the effectiveness of drugs on different species can be used indirectly as phylogenetic evidence?

2 Comparison of analyses of protein structures of different species is lacking.

3 The descriptions in Figs. 1 and 2 are too simple, especially Fig. 1, which is not necessary as part of the article and is not close enough to the content of the paper PFOR, the authors could consider combining the two figures into one.

4 Fig. 3 and Fig. 2 have a very transitive relationship. There is a lack of causality. Not related to the origin and development of PFOR.

5 The chemical formulae in the diagram are out of proportion.

6 Fig. 5, 6, and 7 are similar and do not explain the role and relationship between PFOR and the drug, suggesting a merger.

7 The author mentioned that "Metronidazole exercises its antimicrobial effect by inference with nuclear acid synthesis results in DNA damage in the target pathogens. toxic radial metabolites". But,PFOR activation of metronidazole may not be the main mechanism.

8 Fig. 13 is not clear. fig. 13, 14 whether the scale is drawn wrongly.

9 Line 677 figure 27 should be figure 22. Line 704 figure 27 is incorrect.

10 The hypothesis that pfor is likely to be horizontally obtained is also not supported by further evidence in this manuscript.

11 Metronidazole did not be considered as a potential drug candidate for the treatment of infections caused by Cryptosporidium.

12 Whether the title of the manuscript is too lengthy without highlighting the topic?

Reviewer 2 Report

Comments and Suggestions for Authors

Comments: 

Authors performed the bioinformatics analysis of PFOR evolution relationships between bacteria and protozoan. Multiple techniques, search as phylogeny tree, BLAST search, HMM profilings and GO analysis were applied. The results well supported the aim of the studies, while there are a few areas that need to be improved.

1. Although there are more than 20 figures, most of them were not mentioned in the main text, making it hard to read and understand the story. Besides, authors should spend time combining multiple figures so that it will be easier to interpret the results. Figure 5 & 6 can be combined in one figure talking about the mechanism of pro-drug activation by PFOR. 

2.The label of sections was off. For instance, there was only one section, 2.1, under section 2. There is no need to label 2.1 as the sub-section. 3.1.1 appeared before section 3.  Authors must be careful to rearrange the section labeling. 

3. Figure 5~8 need to be re-draw using chemical structure drawing software such as ChemDraw. As a biochemist majoring in natural product chemistry, it's not accepted to simly copy-paste chemical structures from different sources. 

4. There are two pro-drugs showing the chemical structure. Authors should re-draw the structure. Meanwhile, they should be combined with some other chemical reaction schemes, such as Figure 8, so that it won't be a sole chemical structure.

5. Figure 9~12 are drawn using EXCEL software, it is better to combine them in one figure and I don't understand the reason for having glowing style of the bar plot and line plot.

6. There are several figures, such as Figure 15, Figure 24, 25, which were scaled improperly and the font within the figure were stretched improperly. Authors should always re-scale the figure by fixing its width/length ratio. 

Reviewer 3 Report

Comments and Suggestions for Authors

The distribution of the typical bacterial redox enzyme pyruvate-ferredoxin oxidoreductase (PFOR) in eukaryotes remains intriguingly puzzling. Free-living mitochondriate protist-like photosynthetic algae also express homologs of PFOR. This study analyzes the distribution of PFOR in 30 eukaryotes, with a special focus on pathogenic protozoa. The work is interesting and provides novel insights into bacterial PFOR. The results also have the potential for practical applications in production. For the most part, this work is based on sound designs. The manuscript can be further improved by considering the following points. 

Major comments

The format of the paper is somewhat unusual. For instance, I did not come across an abstract section. Instead, there is a summary that extends to several hundred words. However, the instructions for this journal explicitly state, "The abstract should be a total of about 200 words maximum.

Minor comments

1. L36 Please providing a brief context on the significance of understanding the localization of PFOR in protozoa and its potential origins could enhance the introduction.

2. L83 Expand a bit on "maintenance transfer" and its significance. Why does the maintenance of pre-existing functions matter, and how does it impact the evolutionary process?

3. Fig. 1 and Fig. 2. The content presented in this figure is too superficial; it should focus more on the distribution of PFOR in different species and its transfer patterns.

4. Table 2 I'm not sure about the connection between the content in this table and pyruvate-ferredoxin oxidoreductase (PFOR).

5. L320 Is Section 3 after Section 3.1.2?
